# Herring Milt Protein Hydrolysate Improves Insulin Resistance in High-Fat-Diet-Induced Obese Male C57BL/6J Mice

**DOI:** 10.3390/md17080456

**Published:** 2019-08-03

**Authors:** Yanwen Wang, Jacques Gagnon, Sandhya Nair, Shelly Sha

**Affiliations:** 1Aquatic and Crop Resource Development Research Center, National Research Council of Canada, Charlottetown, PE C1A 4P3, Canada; 2Department of Biomedical Sciences, University of Prince Edward Island, Charlottetown, PE C1A 4P3, Canada; 3VALORēS Research Institute Inc., Shippagan, NB E8S 1J2, Canada; 4Campus of Shippagan, University of Moncton, Shippagan, NB E8S 1P6, Canada

**Keywords:** herring milt protein hydrolysate, type 2 diabetes, diet-induced obese mice, blood glucose, insulin, leptin, oral glucose tolerance, HOMA-IR, HOMA-β

## Abstract

Protein consumption influences glucose homeostasis, but the effect depends on the type and origin of proteins ingested. The present study was designed to determine the effect of herring milt protein hydrolysate (HPH) on insulin function and glucose metabolism in a mouse model of diet-induced obesity. Male C57BL/6J mice were pretreated with a low-fat diet or a high-fat diet for 6 weeks. Mice on the high-fat diet were divided into four groups where one group continued on the high-fat diet and the other three groups were fed a modified high-fat diet where 15%, 35%, and 70%, respectively, of casein was replaced with an equal percentage of protein derived from HPH. After 10 weeks, mice that continued on the high-fat diet showed significant increases in body weight, blood glucose, insulin, and leptin levels and exhibited impaired oral glucose tolerance, insulin resistance, and pancreatic β-cell dysfunction. Compared to mice fed the high-fat diet, the 70% replacement of dietary casein with HPH protein reduced body weight, semi-fasting blood glucose, fasting blood glucose, insulin, leptin, and cholesterol levels and improved glucose tolerance, homeostasis model assessment of insulin resistance (HOMA-IR), and homeostasis model assessment of β-cell function (HOMA-β) indices. The 35% replacement of dietary casein with HPH protein showed moderate effects, while the 15% replacement of dietary casein with HPH protein had no effects. This is the first study demonstrating that replacing dietary casein with the same amount of protein derived from HPH can prevent high-fat-diet-induced obesity and insulin resistance.

## 1. Introduction

Type 2 diabetes mellitus (T2DM) is at present one of the most common human disease conditions, and its global prevalence is expected to continue to increase at an alarming pace [1]. In 2017, it was estimated that 451 million people (aged between 18 and 99 years) had diabetes, and this is predicted to increase to 693 million by 2045 [2]. Approximately 5 million deaths in adults (age 20–99) were deemed to be attributable to diabetes and had an associated healthcare expenditure of 850 billion USD in 2017 [2]. It is evident that diabetes has become a worldwide epidemic, spreading from industrialized nations to the emerging economies of Asia, Latin America, and Africa. It is now a serious threat to life quality and imposes a heavy burden on health care systems and on the global economy. Although T2DM can be treated and many therapeutic drugs are presently available, the long-term effectiveness and adverse effects of such approaches raise challenges. In fact, T2DM is described as a reversible metabolic state that, if diagnosed at a pre-diabetic or early diabetic stage, can be treated by changing dietary habits and lifestyle [3,4].

However, development of T2DM is multi-faceted, and although genetic factors pay a part, obesity and its correlates, including insulin resistance and loss of β-cell mass and/or function, have been considered the major risk factors [5,6]. In line with this notion, it is well known that high levels of dietary carbohydrates and/or fat increase the risk for developing T2DM, while the effects of diets rich in protein are controversial [7,8]. The effect of protein on diabetes is dependent on the type and origin of proteins ingested. Animal proteins, especially red meat proteins, may increase the risk for developing T2DM, while plant proteins tend to be beneficial [9,10]. If red meat is used as the protein source, a high intake of animal proteins may result in a concomitant high consumption of fat as well, particularly saturated fatty acids that have been proven to increase the risk for developing metabolic diseases [11]. Nevertheless, several studies have shown the beneficial effects of fish proteins; for example, cod protein was reported to improve glucose metabolism, glucose tolerance, and insulin sensitivity [8,12,13].

Fish milt is different from fish meat and has been treated as a waste product in fish processing plants or used as a low-grade product. However, fish milt contains a high content of proteins and polyunsaturated fatty acids (PUFA), predominantly omega-3 eicosapentaenoid acid (EPA) and docosahexaenoic acid (DHA) that are mainly incorporated into phospholipids [14]. Due to its high nutritional values, herring milt is used as a food additive for malnourished children in some developing countries [15]. With a worldwide increase of protein consumption due to the increase of the population and dietary preferences, there is an increasing global demand for high-quality protein products. Atlantic herring provides over half of all herring supplied to the global marketplace, and accordingly, a large amount of herring milt is generated during processing. Limited information is available pertaining to the use and potential benefits of herring milt protein for health and wellness purposes, with one study published on the effects of supplementation of herring roe and herring milt in mice in 2012 [15]. The objective of the present study was to test and determine the effect of herring milt protein hydrolysate (HPH) on insulin resistance in mice with high-fat-diet-induced obesity—a well-established animal model of human insulin resistance and T2DM [16].

## 2. Results

### 2.1. Chemical Composition of HPH

Chemical analysis revealed that the HPH contained 70.8% crude protein, 10.8% fat, 12.7% ash, and 6.6% moisture. Further analysis showed HPH to have a high nutritional value and a fatty acid profile of 42.0% saturated, 34.7% monounsaturated, and 23.3% polyunsaturated fatty acids. The PUFA fraction had an omega-3-to-omega-6 ratio of 10.7 and the omega-3 PUFA were predominantly EPA (8.0%) and DHA (10.8%). The amino acid composition of HPH had a very high level of arginine (11.4%) that alone amounted to 29% of the total amino acid fraction, which is consistent with a previously published report [17].

### 2.2. Body Weight and Food Intake

During the first two weeks, the food intake was similar between the low-fat diet control (LFC) and high-fat diet control (HFC) groups. Starting the third week, the food intake in the HFC group became lower (*p* < 0.05), and this effect remained at about the same lowered level for measurements made in Weeks 5, 6, and 8 (Table 1). Nevertheless, the body weight of the HFC group was higher (*p* < 0.05) throughout the study and consistently increased over the treatment period (Table 2). There were no differences observed in body weight or food intake amongst the three treatment groups (*p* > 0.05). The initial body weight was similar for the HFC and the three treatment groups. However, mice fed a modified high-fat diet where 70% casein was replaced with the equivalent amount of protein derived from HPH (HPH70) started losing weight in Weeks 1 and 2 and became significantly lighter (*p* < 0.05) after 3 weeks of treatment as compared to those fed the HFC diet, and this effect remained until the completion of the 10-week study period. No reductions in body weight were observed in mice fed a modified high-fat diet where 35% casein was replaced with the equivalent amount of protein derived from HPH (HPH35) or a modified high-fat diet with 15% casein being replaced by the same amount of protein derived from HPH (HPH15) as compared to the HFC group.

### 2.3. Blood Glucose

The HFC group showed higher (*p* < 0.05) semi-fasting (4–6 h) glucose levels during the entire study period relative to the LFC group (Figure 1). As compared to the HFC group, the treatment groups HPH70 and HPH35 showed lower (*p* < 0.05) blood glucose levels after 2 weeks of treatment and again after 5 weeks. This effect continued to be observed in the HPH70 group until the end of the study. HPH15 did not show a significant effect on the semi-fasting blood glucose as compared to the HFC diet. Fasting (12 h) blood glucose was higher (*p* < 0.0001) in the HFC group than in the LFC group (Table 3) but reversed (*p* < 0.001) in the HPH70 group. The HPH15 and HPH35 groups did not show significant effects as compared to the HFC group.

### 2.4. Oral Glucose Tolerance

After 4 weeks of treatment (Week 5), the HFC group showed impaired glucose tolerance as evidenced by having higher (*p* < 0.05) blood glucose levels than the LFC group at every time point during the oral glucose tolerance test (OGTT) (Figure 2A). The treatment groups did not show a statistically significant effect, although there were trends towards a lowering of blood glucose. The HPH15 and HPH35 groups showed similar effects, and HPH70 appeared to have a stronger effect. The OGTT was repeated during Week 8 of treatment. Similar to the data obtained during Week 5, the HFC group consistently had higher glucose levels than the LFC group at the beginning of the OGTT and at each time point after glucose loading (Figure 2B). All treatment groups tended to have lower glucose levels than the HFC group. Again, the glucose tolerance curves for HPH15 and HPH35 were similar and the HPH70 had consistently lower levels than the HPH15 and HPH35 groups. However, the differences were not statistically significant, which might be a result of large variations observed within groups. When the effect was assessed using the area under the curve (AUC) values, the HFC showed higher values in both Week 5 (*p* < 0.0001) and Week 8 (*p* < 0.001) than the LFC (Figure 3). The HPH70 treatment showed a significantly lower AUC than the HFC group at both Week 5 (*p* < 0.05) and Week 8 (*p* < 0.01) time points. The AUC values for HPH35 (*p* = 0.097) and HPH15 (*p* = 0.063) showed marginal differences when compared to the HFC group.

### 2.5. Fasting Serum Insulin, Leptin, Adiponectin, and Free Fatty Acids

As shown in Table 3, the HFC group had a fasting serum insulin level 2.7-fold higher than that of the LFC group (*p* < 0.0001). HPH70 reduced (*p* < 0.01) fasting serum insulin by 50% when compared to the HFC group, while no significant difference was observed in either the HPH15 or HPH35 group. Similar effects were also observed for the serum leptin levels, where the HFC group had an over 1.4-fold increase (*p* < 0.0001) in fasting serum leptin levels compared to the LFC group. The HPH70 group showed a 40% reduction (*p* < 0.01) in fasting serum leptin as compared to the HFC group. There were no significant effects in fasting leptin levels noted for the HPH15 or the HPH35 group. Serum adiponectin was lowered (*p* < 0.05) in the HFC group as compared to the LFC group, and no treatment effects were observed. The serum free fatty acids did not show any significant differences between the HFC and LFC groups or among the HFC and three treatment groups.

### 2.6. HOMA-IR and HOMA-β

The homeostasis model assessment of insulin resistance (HOMA-IR) index was 4.7-fold higher (*p* < 0.0001) in the HFC group than in the LFC group (Table 3). The treatment showed a significant effect where the HPH70 group had a 64% decrease (*p* < 0.01) in this index when compared to the HFC group. The HPH35 group showed a marginal difference (*p* = 0.07), while HPH15 did not show a significant effect. There was a marked difference (*p* < 0.05) in the homeostasis model assessment of β-cell function (HOMA-β) between the HFC and LFC groups, with the index being 87% lower (*p* < 0.0001) in the HFC as compared to the LFC group. The HPH15, HPH35, and HPH70 treatments dose-dependently increased HOMA-β: it was sixfold higher in the HPH70 group (*p* < 0.001) and doubled in the HPH35 group (*p* < 0.05) as compared to in the HFC group, but no statistical difference was observed for the HPH15 group.

### 2.7. Fasting Serum Total Cholesterol and Triglycerides

As shown in Table 3, the high-fat diet induced hypercholesterolemia in mice, resulting in a higher (*p* < 0.0001) level of serum total cholesterol in the HFC as compared to the LFC group. By contrast, fasting serum triglyceride levels were lower (*p* < 0.01) in the HFC group as compared to the LFC group. HPH70 reduced serum total cholesterol by 13% (*p* < 0.05) but had no effect on triglyceride levels when compared to the HFC group. HPH15 and HPH35 treatments did not show any significant effects on either cholesterol or triglyceride levels when compared to the HFC group.

## 3. Discussion

Feeding a high-fat diet induced significant effects in mice in terms of insulin resistance, impaired oral glucose tolerance, and elevation of blood cholesterol that are well in a line with the results of previous studies conducted in mice and other animal models [16,18]. Interestingly, when 70% dietary casein was replaced with an equivalent amount of proteins derived from HPH, insulin resistance was significantly attenuated as early as 2 weeks into the treatment. Moderate effects were observed in mice treated with HPH35. The present study demonstrated for the first time that HPH improved glucose homeostasis in a mouse model of high-fat-diet-induced obesity and insulin resistance.

Chronic consumption of a high-fat diet induces obesity, hyperinsulinemia, hyperglycemia, and insulin resistance in C57BL/6J mice [18]. Increased weight gain and induction of insulin resistance by a high-fat diet were also observed in other rodent species [19]. In the present study, mice on the high-fat diet consumed less food but higher energy as compared to the low-fat controls, resulting in a significant increase in weight gain and body weight, in accordance with many previous studies [18,20]. Interestingly, consumption of the HPH70 diet caused a significant reduction in weight gain as compared to the HFC group. As the diets were isocaloric and the food intake tended to be higher than that in the HFC group, the cause of weight reduction in the HPH70 group might be a result of a decrease in energy absorption and utilization and/or an increase in energy expenditure. Indeed, it has been reported that dietary supplementation of herring milt enhances hepatic fatty acid beta-oxidation and reduces fatty acid synthase activity in mice [15]. Another study has shown that consumption of fish protein decreases fat absorption [21]. It has also been reported that fish protein hydrolysates increase hepatic fatty acid oxidation and alter body fatty acid composition [22]. The HPH used in this study contained over 10% fat of which a significant fraction was omega-3 EPA and DHA, which have been shown to stimulate lipid oxidation and increase metabolic rate, resulting in the reduction of fat mass [23,24].

In a diet-induced obese mouse model, overnight fasting every week or two weeks ruins the model and makes the blood glucose level return to normal (unpublished data in another study). Therefore, semi-fasting blood glucose is commonly used to monitor the change in blood glucose levels during the course of treatment, and fasting blood glucose is measured at the end of the study [18,19]. In agreement with the observed weight changes, mice in the HFC group showed significantly higher semi-fasting blood glucose during the entire 10-week study period. Mice in the HPH35 group tended to have lower blood glucose levels where semi-fasting blood glucose was found to be lower at 2 and 5 weeks of treatment, even though there were no significant effects on either body weight or food intake. Further reductions in semi-fasting blood glucose were seen in mice of the HPH70 group. A similar pattern of results was observed in fasting blood glucose levels. A significant correlation was found between the body weight and fasting blood glucose (r = 0.9489, *p* < 0.0001). Although no significant effects on the body weight were seen in the HPH35 and HPH15 groups, the significant correlation between body weight and fasting blood glucose may suggest that the reduction of blood glucose in mice treated with HPH might be a confounded result of weight reduction and other beneficial effects induced by HPH. Similar results were reported for other marine protein sources such as cod–scallop, which reduced weight gain and serum glucose when used as the dietary protein source in mice fed a high-fat diet [25]. An increase in semi-fasting and fasting blood glucose levels is indicative of insulin resistance, which is commonly assessed by performing glucose tolerance testing and measuring a number of biomarkers in blood samples [18,19]. Mice in the HFC group showed significantly higher blood glucose levels at every time point of the two OGTTs conducted when compared to mice in the LFC group, demonstrating impaired glucose tolerance, in agreement with the results of many previous studies [19,26,27]. Mice fed HPH displayed consistent reductions in blood glucose level at each time point during each OGTT when compared to mice fed the high-fat diet, and a significantly lower AUC was seen in mice of the HPH70 group. HPH70 markedly lowered fasting serum insulin, suggesting a protective effect against the development of insulin resistance in obese and insulin-resistant mice. This notion was further supported by a reduction in the HOMA-IR index in the HPH70 group. Similar to our results, several other studies have shown that dietary fish proteins significantly influence insulin sensitivity and glucose homeostasis [8,13,28]. Although not examined in the present study, the published work suggests that fish protein beneficially alters metabolic phenotypes and processes related to insulin signaling. A study in rats with fructose-induced metabolic syndrome showed that dietary sardine protein lowered inflammation and oxidative stress [12], which are well-established factors negatively influencing insulin sensitivity. Another study in humans showed improvement of insulin sensitivity by dietary cod protein [13]. The improved insulin sensitivity by HPH in the present study might be a result, at least in part, of the enhancement of the insulin signaling pathway as observed in high-fat-fed obese rats where cod protein restored the insulin-induced activation of phosphatidylinositol 3-kinase/Akt and GLUT4-translocation [8].

Pancreatic β-cell function is another important parameter used to assess the benefits of a compound or treatment on glucose metabolism and glucose homeostasis. In the present study, mice in both the HPH70 and HPH35 groups showed a marked increase in their HOMA-β index, demonstrating that HPH may have a protective effect on pancreatic β-cells, resulting in improved β-cell function in obese and insulin-resistant mice. The observed benefits of fish protein on β-cell function could be a result of several nutrients and bioactives in HPH [29]. A recent study showed that fish-derived peptides beneficially influenced pathways involved in controlling body composition, lipid profiles, and the regulation of glucose metabolism [30]. A critical review suggested that unsaturated PUFA (especially omega-3) are beneficial to the mass and function of pancreatic β-cells though inhibiting inflammatory pathways and regulating cell proliferation and apoptosis [31]. Fish proteins have high levels of taurine, which is reported to help ameliorate hyperglycemia and dyslipidemia by reducing insulin resistance and leptin levels [32,33,34]. Taurine is also reported to have an effect on insulin release from pancreatic β-cells [35]. Moreover, HPH contains a very high level of L-arginine, which has been demonstrated to potentiate glucose-induced insulin secretion mediated by membrane depolarization, which stimulates insulin secretion through protein kinase A- and C-sensitive mechanisms [36]. A further study showed that L-arginine induced insulin release through regulating G proteins, particularly the isoform Gα_i2_ [37].

Leptin increases energy expenditure by enhancing systemic and brown adipose glucose utilization. The increase of leptin levels in obese mice is indicative of leptin resistance and vice versa [38]. Similar to the previous study, increased leptin levels and leptin resistance were observed in the current study in mice fed the high-fat diet. It was reported that L-arginine can mediate leptin effects [39], suggesting that it is possible that the consumption of HPH replacing dietary casein improved leptin resistance through L-arginine action and thereby increased glucose utilization and energy expenditure and decreased body weight. A similar study reported that sardine protein lowered plasma leptin and glucose levels and improved impaired glucose tolerance [12]. The reduction of serum leptin levels by HPH might be a consequence of improved leptin resistance or a confounded effect of improving both leptin and insulin resistances. All these phenotypes affect body weight and composition. On the other hand, it is not possible to exclude the possibility that HPH affected energy metabolism and weight gain as discussed earlier, resulting in the improvement of leptin and insulin resistance.

In the present study, blood adiponectin and free fatty acid levels in mice were not affected by HPH supplementation, while body weight and most metabolic variables were significantly improved. Previous studies have shown mixed results regarding the association between blood adiponectin levels and body weight and other metabolic phenotypes [18,19,32]. In a study, dietary supplementation of taurine, a high-content component in fish protein, improved insulin resistance and dyslipidemia and reduced leptin levels but had no effect on adiponectin levels [32]. Similarly, significant weight reduction and improvement of metabolic phenotypes were induced by dietary supplementation of flaxseed lignans; however, there was no effect on the blood adiponectin levels even though there was a significant reduction of blood free fatty acid levels [18], which changed reciprocally with adiponectin levels in another study using shrimp oil [19]. Adipose dysfunction in obesity and diabetes induces lipolysis and increases circulating free fatty acids to promote ectopic fat deposits [40]. Lipid accumulation in the liver and muscle is generally accepted as the main cause of insulin resistance [41,42,43]. Lipolysis-induced lipid accumulation in the liver attenuates insulin sensitivity and increases hepatic gluconeogenesis, responsible for hyperglycemia during fasting in diabetes [44]. Adiponectin is secreted by adipose tissue, and its level is a marker of the total triglyceride lipolytic rate per adipose tissue mass [45]. Adiponectin also enhances adipocyte lipid storage [46], preventing ectopic lipid accumulations. The present study used a diet-induced obese mouse model, and the blood concentrations of free fatty acids or adiponectin were not significantly affected by HPH and not hugely different between the HFC and LFC either. This may suggest that insulin resistance was not severe or the time was not sufficient to cause significant adipose dysfunction [45,46]. It is also possible that HPH improved body weight and most metabolic variables independent of the adipose tissue, such as through effects on the insulin signaling pathway, pancreatic β-cell mass and function, intestinal fat absorption, and hepatic fat oxidation that were discussed earlier.

The result of the current study also revealed that feeding mice with a high-fat diet increased their blood total cholesterol level. Interestingly, mice in the HPH70 group showed a significantly lower level of blood cholesterol when compared with mice in the HFC group. This observation is in agreement with the reported hypocholesterolemic effect of protein hydrolysates from salmon flesh remnants in Wistar and genetically obese Zucker (fa/fa) rats [47]. The effect could be a result of several mechanisms, including decreased cholesterol absorption, inhibited cholesterol biosynthesis, and/or increased cholesterol catabolism into bile and secretion together with bile from the liver into the intestine for excretion. Several studies have shown that fish protein or protein hydrolysate reduces serum cholesterol levels by decreasing intestinal absorption and increasing fecal cholesterol and bile acid excretions [47,48]. It has also been reported that fish protein hydrolysates reduce the activity of acyl-CoA:cholesterol acyltransferase, an enzyme that is required for the conversion of free cholesterol to cholesterol ester in the enterocyte following absorption and for cholesterol to be secreted out of the basolateral side of the enterocyte into the lymphatic system and then into the blood stream [47]. Moreover, fish protein decreases the solubility of cholesterol in micelles, which transfer cholesterol in the small intestine across the unstirred water layer to the epithelium for absorption [48]. A reduction of micellar solubility of cholesterol in the intestinal lumen or cholesterol esterification following entry into the enterocyte decreases cholesterol absorption. The reduction of blood triglyceride levels in mice of the HFC group is consistent with previous reports [49], and HPH did not show a significant effect.

In conclusion, the replacement of casein protein with HPH in mice with obesity and insulin resistance induced by a high-fat diet resulted in significant improvements to semi-fasting and fasting blood glucose levels, oral glucose tolerance, insulin resistance, and β-cell function, accompanied by reductions in blood insulin, leptin, and cholesterol levels. The results suggest that HPH possesses potential for preventing or treating diet-induced obesity and metabolic complications. As weight reduction was also observed in mice treated with HPH, it is not possible to conclude that the effects were a result of weight reduction or perhaps a confounded effect of the benefits associated with weight reduction and independent improvements in terms of insulin secretion, insulin sensitivity, and leptin sensitivity. Further studies are required to understand the underlying mechanisms, especially those that are independent of weight reduction, and further identify the responsible bioactive components in HPH such as peptides and amino acids.

## 4. Materials and Methods

### 4.1. Preparation of HPH

Herring milt was obtained from a herring fish processing plant in Lameque, NB, Canada and kept at −20 °C until further processing. An aliquot of 2.27 kg of herring milt was blended in a 4 L Waring LBC15 blender (Torrington, CT, USA). The blended mixture was transferred into a 6 L Bioflo115 bioreactor (New Brunswick Scientific, Enfield, CT, USA) with the addition of 1.25 L of distilled water. The mixture was heated up to 50 °C while agitating at 400 rpm and maintaining the pH at 8.0. Then, 12.5 g of protamex (18.74 U) (Novozymes, Franklington, NC, USA) and 15.6 g of alcalase (37.44 U) (Novozymes, Franklington, NC, USA) were added. After 6 h of incubation, the enzymes were inactivated by heating the mixture at 85 °C for 5–10 min. The hydrolysed mixture was dried in a freeze-dryer (LabConco, Kansas City, MO, USA). The dried material was homogenized using a blender and then placed in a vacuum plastic pack.

### 4.2. Analysis of HPH

Unless otherwise specified, proximate analyses were performed following the AOAC methods (Association of Official Analytical Chemists, 2012). The ash content was determined by placing the sample in a muffle furnace at 550 °C for 5 h (AOAC 920.153), while the moisture content was determined by drying the sample in an air-forced oven at 135 °C for 2 h (AOAC 930.15). Total nitrogen was analyzed using the combustion method (AOAC 992.15), and the content of crude lipids was determined using the Folch method [50].

### 4.3. Animals and Diets

Forty-eight male C57BL/6J mice fed a high-fat diet (60% energy from fat) starting at the age of 5 weeks and 12 male C57BL/6J mice fed a low-fat diet all the time were purchased from Jackson Laboratories (Jackson, FL, USA) at the age of 10 weeks. After arrival, they were housed individually in regular mouse cages with a 12:12 h dark/light cycle and free access to water and the same high-fat (D12492) and low-fat (D12450B) diets (Research Diets Inc., New Brunswick, NJ, USA), respectively. After a week of acclimation, mice on the high-fat diet were weighed and randomly divided into four groups. One group continued on the high-fat diet and was used as the HFC, and the other three were fed a modified high-fat diet where 15%, 35%, or 70% casein was replaced by an equivalent amount of protein derived from HPH. The treatments lasted for 10 weeks. The fat contained in HPH was adjusted by reducing the equivalent amount of fat in the form of lard. The mice fed the low-fat diet were used as the LFC. The diet composition is provided in the Appendix A. Food consumption was monitored daily, and mice were weighed weekly. Semi-fasting (4–6 h) blood glucose was measured every week with a glucometer (ACCU-CHECK Aviva; Roche, Basel, Switzerland) using blood via tail snip. Oral glucose tolerance testing was performed during Weeks 5 and 8 of the treatment. At the end of experiment, mice were fasted overnight, and blood was collected by cardiac puncture after anesthesia with inhalation of isoflurane (Pharmaceutical Partners of Canada Inc., Toronto, ON, Canada) into serum collection tubes. Serum was obtained by centrifugation at 1500× *g* for 20 min and stored in cryogenic vials at −80 °C for later analysis of biomarkers. The animal use and experimental procedures were approved by the Joint Animal Care and Research Ethics Committee of the National Research Council Canada in Charlottetown and the University of Prince Edward Island. The study was conducted in accordance with the guidelines of the Canadian Council on Animal Care.

### 4.4. Oral Glucose Tolerance Test

Glucose tolerance tests were performed by orally administering glucose at a dose of 2 g/kg body weight after 4–6 h fasting. The blood glucose concentration was measured from tail vein blood at 0, 30, 60, 90, and 120 min using an Accu-Chek glucometer (Roche Diagnostics, Toronto, ON, Canada). The area under the curve (AUC) of glucose concentrations was calculated and used as a second parameter to assess oral glucose tolerance.

### 4.5. Serum Insulin, Leptin, and Adiponectin Analysis

Fasting serum insulin, leptin, and adiponectin were analyzed using commercial mouse ELISA kits following the kits’ instructions. The kits were obtained from Crystal Chem (Downers Grove, IL, USA) for insulin, Cayman Chemicals (Ann Arbor, MI, USA) for leptin, and R & D Systems (Minneapolis, MN, USA) for adiponectin.

### 4.6. Homeostasis Model Assessment of Insulin Resistance and β-Cell Function

HOMA-IR was calculated as (glucose in mmol/L × fasting insulin in mU/L)/22.5. HOMA-β was calculated as (20 × insulin in mU/L)/(glucose in mmol/L − 3.5).

### 4.7. Analysis of Serum Lipids

Concentrations of serum total cholesterol and triglycerides were measured using enzymatic methods with reagents purchased from Pointe Scientific Inc. (Canton, MI, USA). Free fatty acids were measured using free fatty acid quantification kits (BioVision, Mountain View, CA, USA) according to the kit instructions.

### 4.8. Analysis of Serum Glucose

Fasting serum glucose concentrations were measured using an enzymatic method. The reagents were purchased from Genzyme Diagnostics (Charlottetown, PE, Canada). The analyses were carried out in accordance with the manufacturer’s instructions.

### 4.9. Statistical Analysis

All data analyses were performed using SAS software version 9.4 (SAS Institute, Cary, NC, USA). The difference between the HFC and LFC was analyzed using Student’s *t*-test while the treatment effects were determined using one-way ANOVA. Repeated measures *t*-test and repeated measures one-way ANOVA were employed for the parameters that were measured multiple times. When a significant treatment effect was detected, differences among the HFC, HPH15, HPH 35, and HPH70 groups were analyzed using the least-squares means test. The significance level was set at *p* < 0.05. The results are presented as means ± SEM.

## Figures and Tables

**Figure 1 marinedrugs-17-00456-f001:**
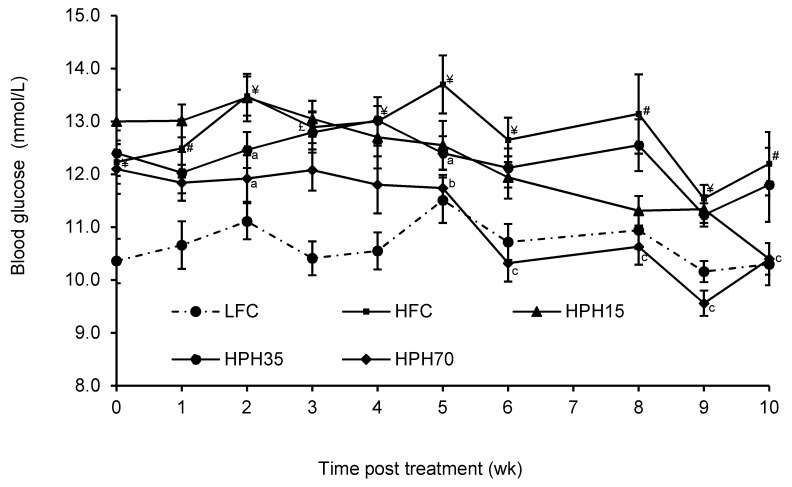
Effect of HPH supplementation on semi-fasting blood glucose concentration in mice fed a high-fat diet. The results are presented as means ± SEM (*n* = 11−12). The difference between the HFC and LFC was analyzed using Student’s *t*-test with repeated measures. The treatment effect was analyzed using one-way ANOVA with repeated measures, and differences among the HFC, HPH15, HPH35, and HPH70 were determined using the least-squares means test. The significance level was set to 0.05. ^#^
*p* < 0.05, ^¥^
*p* < 0.01; ^£^
*p* < 0.001 as compared with LFC; ^a^
*p* < 0.05, ^b^
*p* < 0.01, ^c^
*p* < 0.001 compared with HFC. LFC, low-fat control; HFC, high-fat control; HPH15, HFC diet with 15% of casein being replaced with the same amount of protein derived from herring milt protein hydrolysate; HPH35, HFC diet with 35% of casein being replaced with the same amount of protein derived from herring milt protein hydrolysate; HPH70, HFC diet with 70% casein being replaced with the same amount of protein derived from herring milt protein hydrolysate.

**Figure 2 marinedrugs-17-00456-f002:**
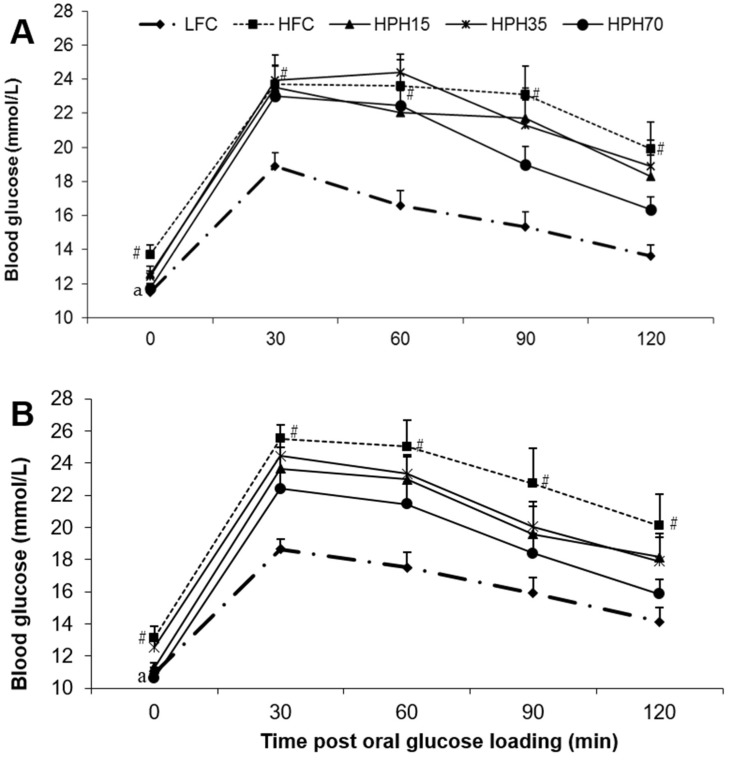
Effect of HPH supplementation on oral glucose tolerance in obese and insulin-resistant mice. The difference between the HFC and LFC was analyzed using repeated-measures *t*-test. The treatment effect was analyzed using one-way ANOVA with repeated measures, and when a significant treatment effect was obtained, differences among HFC, HPH15, HPH35, and HPH70 were determined by pairwise comparisons using the least-squares means test. Data are presented as means ± SEM (*n* = 11−12). (**A**) The result of an oral glucose tolerance test conducted during Week 5 of the treatment. (**B**) The results of an oral glucose tolerance test conducted during Week 8 of the treatment. The significance level was set to 0.05. ^#^
*p* < 0.05 compared with LFC; ^a^
*p* < 0.05 compared with HFC. LFC, low-fat control; HFC, high-fat control; HPH15, HFC diet with 15% of casein being replaced with the same amount of protein derived from herring milt protein hydrolysate; HPH35, HFC diet with 35% of casein being replaced with the same amount of protein derived from herring milt protein hydrolysate; HPH70, HFC diet with 70% casein being replaced with the same amount of protein derived from herring milt protein hydrolysate.

**Figure 3 marinedrugs-17-00456-f003:**
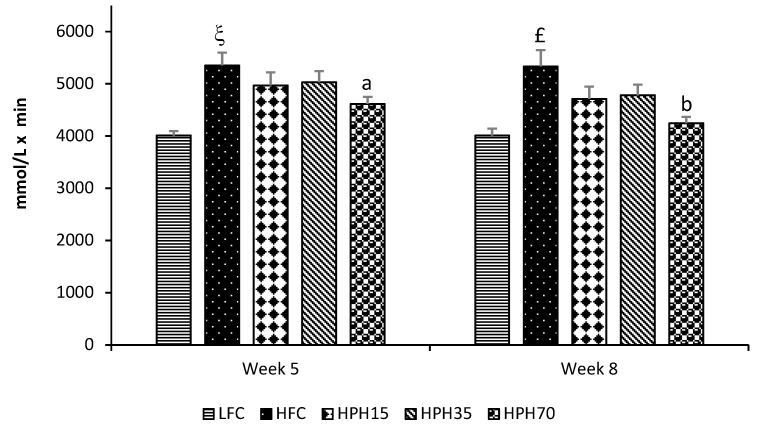
Effect of HPH supplementation on the area under the curve (AUC) of oral glucose tolerance in mice fed a high-fat diet. Difference between the HFC and LFC was analyzed using Student’s *t*-test. The treatment effect was analyzed using one-way ANOVA, and when a significant treatment effect was obtained, differences among HFC, HPH15, HPH 35, and HPH70 were determined by pairwise comparisons using the least-squares means test. Data are presented as means ± SEM (*n* = 11−12). The significance level was set to 0.05. ^£^
*p* < 0.001 and ^ξ^
*p* < 0.0001 compared with LFC; ^a^
*p* < 0.05 and ^b^
*p* < 0.01 compared to HFC. LFC, low-fat control; HFC, high-fat control; HPH15, HFC diet with 15% of casein being replaced with the same amount of protein derived from herring milt protein hydrolysate; HPH35, HFC diet with 35% of casein being replaced with the same amount of protein derived from herring milt protein hydrolysate; HPH70, HFC diet with 70% casein being replaced with the same amount of protein derived from herring milt protein hydrolysate.

**Table 1 marinedrugs-17-00456-t001:** Effect of herring milt protein hydrolysate (HPH) supplementation on the food intake (g) of mice fed a high-fat diet.

Treatment	Time Post Treatment (Week)						
	1	2	3	5	6	8	10
LFC	3.1 ± 0.2	3.0 ± 0.2	3.5 ± 0.1	3.3 ± 0.1	3.8 ± 0.1	3.4 ± 0.2	3.2 ± 0.1
HFC	3.0 ± 0.1	3.0 ± 0.1	2.8 ± 0.1 ^#^	2.9 ± 0.1 ^#^	3.0 ± 0.1 ^#^	2.9 ± 0.1 ^#^	3.0 ± 0.1
HPH15	2.7 ± 0.1	2.9 ± 0.1	2.7 ± 0.1	2.7 ± 0.1	2.9 ± 0.1	2.9 ± 0.1	3.0 ± 0.1
HPH35	2.6 ± 0.1	2.9 ± 0.1	2.9 ± 0.1	2.9 ± 0.1	2.8 ± 0.1	2.9 ± 0.1	3.1 ± 0.1
HPH70	2.4 ± 0.1	3.1 ± 0.1	3.0 ± 0.1	3.4 ± 0.2	3.3 ± 0.2	3.1 ± 0.1	3.8 ± 0.3

The results are presented as means ± SEM (*n* = 11−12). The difference between the HFC and LFC groups was analyzed using Student’s *t*-test with repeated measures. The treatment effect was analyzed using one-way ANOVA with repeated measures, and differences among the HFC, HPH15, HPH35, and HPH70 groups were determined using the least-squares means test. The significance level was set to 0.05. ^#^
*p* < 0.05 compared with LFC. LFC, low-fat control; HFC, high-fat control; HPH15, HFC diet with 15% of casein being replaced with the same amount of protein derived from herring milt protein hydrolysate; HPH35, HFC diet with 35% of casein being replaced with the same amount of protein derived from herring milt protein hydrolysate; HPH70, HFC diet with 70% casein being replaced with the same amount of protein derived from herring milt protein hydrolysate.

**Table 2 marinedrugs-17-00456-t002:** Effect of HPH supplementation on the body weight (g) of mice fed a high-fat diet.

Treatment	Time Post Treatment (Week)									
	0	1	2	3	4	5	6	8	9	10
LFC	34.2 ± 0.5	35.5 ± 0.5	35.7 ± 0.5	35.4 ± 0.5	36 ± 0.6	36.7 ± 0.6	36.7 ± 0.5	37.5 ± 0.7	38.2 ± 0.7	37 ± 0.5
HFC	42.3 ± 1.2 ^#^	43.6 ± 1.3 ^#^	44.8 ± 1.2 ^#^	46.5 ± 1.2 ^#^	47.3 ± 1.2 ^#^	47.6 ± 1.2 ^#^	48.1 ± 1.1 ^#^	49.5 ± 0.9 ^#^	49.8 ± 0.8 ^#^	49.4 ± 0.7 ^#^
HPH15	43.8 ± 1.4	45.0 ± 1.3	45.7 ± 1.1	47.2 ± 1.0	47.8 ± 0.9	48.6 ± 0.8	49.1 ± 0.6	50.2 ± 0.5	50.5 ± 0.6	49.2 ± 1.2
HPH35	43.0 ± 1.1	42.8 ± 1.0	43.3 ± 1.0	45.0 ± 0.9	45.7 ± 0.9	46.4 ± 0.9	45.6 ± 1.1	47.8 ± 0.9	48.1 ± 0.9	48.1 ± 0.8
HPH70	43.1 ± 1.2	41.3 ± 1.2	40.1 ± 1.3	40.4 ± 1.3 ^a^	40.3 ± 1.3 ^b^	41.2 ± 1.4 ^b^	41.3 ± 1.4 ^b^	41.9 ± 1.5 ^b^	40.8 ± 1.5 ^b^	41.1 ± 1.6 ^b^

The results are presented as means ± SEM (*n* = 11−12). The difference between the HFC and LFC was analyzed using Student’s *t*-test with repeated measures. The treatment effect was analyzed using one-way ANOVA with repeated measures, and differences among the HFC, HPH15, HPH35, and HPH70 were determined using the least-squares means test. The significance level was set to 0.05. ^#^
*p* < 0.0001 compared with LFC; ^a^
*p* < 0.05, ^b^
*p* < 0.0001 compared with HFC. LFC, low-fat control; HFC, high-fat control; HPH15, HFC diet with 15% of casein being replaced with the same amount of protein derived from herring milt protein hydrolysate; HPH35, HFC diet with 35% of casein being replaced with the same amount of protein derived from herring milt protein hydrolysate; HPH70, HFC diet with 70% casein being replaced with the same amount of protein derived from herring milt protein hydrolysate.

**Table 3 marinedrugs-17-00456-t003:** Effect of HPH supplementation on serum fasting glucose, insulin, leptin, adiponectin, lipids, β-cell function index, and insulin resistance index in mice fed a high-fat diet.

	LFC	HFC	HPH15	HPH35	HPH70
Fasting blood glucose (mmol/L)	7.88 ± 0.42	10.57 ± 0.27 ^ξ^	9.51 ± 0.54	10.01 ± 0.35	8.37 ± 0.50 ^c^
Serum insulin (ng/mL)	0.45 ± 0.07	1.68 ± 0.20 ^ξ^	1.97 ± 0.38	1.21 ± 0.28	0.84 ± 0.18 ^b^
Serum leptin (ng/mL)	25.80 ± 2.44	60.77 ± 3.08 ^ξ^	56.97 ± 6.17	56.37 ± 2.88	36.70 ± 6.14 ^b^
Serum adiponectin (ng/mL)	10.66 ± 0.31	9.79 ± 0.26 ^#^	8.84 ± 0.36	9.95 ± 0.22	8.99 ± 0.40
Total cholesterol (mg/dL)	160.20 ± 5.95	232.51 ± 8.3 ^ξ^	263.24 ± 17.2	251.70 ± 9.2	202.82 ± 16.8 ^a^
Triglycerides (mg/dL)	60.70 ± 2.18	50.48 ± 2.85 ^¥^	58.24 ± 3.51	61.73 ± 2.18	58.04 ± 5.07
FFA (mmol/mL)	0.13 ± 0.01	0.10 ± 0.01	0.12 ± 0.01	0.11 ± 0.01	0.09 ± 0.01
HOMA-IR	5.53 ± 0.93	31.51 ± 3.41 ^ξ^	34.37 ± 6.08	18.75 ± 4.92	11.27 ± 2.61 ^b^
HOMA-β	0.30 ± 0.06	0.04 ± 0.00 ^ξ^	0.05 ± 0.00	0.09 ± 0.01^a^	0.29 ± 0.04 ^c^

The results are presented as means ± SEM (*n* = 11−12). The difference between the HFC and LFC was analyzed using Student’s *t*-test. The treatment effect was analyzed using one-way ANOVA, and differences among the HFC, HPH15, HPH35, and HPH70 were determined using the least-squares means test. The significance level was set to 0.05. ^#^
*p* < 0.05, ^¥^ p < 0.01; ^ξ^
*p* < 0.0001 compared with LFC; ^a^
*p* < 0.05, ^b^
*p* < 0.01, ^c^
*p* < 0.001 compared to HFC. FFA, free fatty acids; HOMA-IR, homeostasis model assessment of insulin resistance; HOMA-β, homeostasis model assessment of β-cell function; LFC, low-fat control; HFC, high-fat control; HPH15, HFC diet with 15% of casein being replaced with the same amount of protein derived from herring milt protein hydrolysate; HPH35, HFC diet with 35% of casein being replaced with the same amount of protein derived from herring milt protein hydrolysate; HPH70, HFC diet with 70% casein being replaced with the same amount of protein derived from herring milt protein hydrolysate.

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
