# Peer review of "Herring Milt Protein Hydrolysate Improves Insulin Resistance in High-Fat-Diet-Induced Obese Male C57BL/6J Mice"

_marinedrugs, 2019, doi:10.3390/md17080456_

Round 1

Reviewer 1 Report

The manuscript by Prof. Wang, Prof. Gagnon, and his colleagues is a comprehensive description of how the herring milt hydrolysate improves insulin resistance in models of obese mice. The manuscript is well written and described. The studies have utilized very simple methods and the results are well understood. Although a mechanism for these effects remains to be identified, the results are relevant. 

Specific comments

1. Was wondering why the authors measured blood glucose in semi-fasting condition. You will have to explain why you have chosen this condition. and also you used 12hr fasting too. What is the difference you wanted to see? Why not random glucose?

2. Some results are shown in the table but it will be much clear if you show it with line graphs or bar graphs, for example, blood glucose changes by the time. Commonly, blood glucose has shown with the unit of mmol/dL. Please change it. 

3. OGTT results, haven't you done before you started the treatments? You will have to show it as a negative control together with 4weeks and 12 weeks of treatment even it shows the same as 4 weeks or so. Also, please insert one more graph next to it, the AUC as a bar graph. 

4. The authors saw the effect of body weight changes after the treatment. Could you please also discuss its possible effects on fat metabolism? And also you said it might protect the b-cell function. You can simply see the b-cell mass changes before and after treatments with herring milt protein hydrolysate to strengthen your theory.

5. minor comments: please mention the full name at first writing. LFC, HFC, PUFA, EPA, DHA. etc. 

Reviewer 2 Report

The aim of this study was to examine the effect of herring milt protein hydrolysate on insulin sensitivity/resistance in high fat diet-fed mice. The replacement of 70% of casein by herring milt protein hydrolysate reduced body weight gain and improved glucose tolerance, lowered fasting blood insulin and leptin levels as well as improved HOMA-IR and HOMA-beta values. Serum adiponectin was lowered by high fat feeding but was not affected by treatment. The topic and the results are of interest. However, there are also some important concerns to be addressed. 1) The results of this study are mostly descriptive and the mechanisms of the effects observed have not been clarified. It would be of interest to examine the effect of supplementation on some components of the insulin signaling pathway in key insulin-sensitive tissues such as skeletal muscles, liver and adipose tissue. 2) The effect on HOMA-beta suggests that herring proteins improved beta-cell function and insulin secretion. This issue including the putative mechanisms should be discussed. 3) Herring milt protein hydrolysate improved metabolic parameters but also reduced body weight. It is unclear how much of the effect was mediated by reduction of body weight and how much was independent of it. It would be of interest to include additional pair-fed group with food intake restricted to match its values in milt hydrolysate-fed groups to address this issue. 4) What form of adiponectin (total? HMW?) was measured? 5) Milt protein hydrolysate improved body weight and most metabolic variables but had no effect on adiponectin, why?

Round 2

Reviewer 2 Report

The manuscript has been revised according to the reviewers' comments.